# Optimization of Biomass Fuel Composition for Rubber Glove Manufacturing in Thailand

Laddawan Thep-On [1], Shahariar Chowdhury [1,2], Kua-Anan Taechato [1,2], Anil Kumar [3] and Issara Chanakaewsomboon [1,4,*]





1. Faculty of Environmental Management, Prince of Songkla University, Songkhla 90110, Thailand
2. Environmental Assessment and Technology for Hazardous Waste Management Research Center, Faculty of Environmental Management, Prince of Songkla University, Songkhla 90110, Thailand
3. Centre for Energy and Environment, Delhi Technological University, Delhi 110042, India
4. The Sustainable Innovation Center (SIC-PSU), Faculty of Environmental Management, Prince of Songkla University, Songkhla 90110, Thailand
* Correspondence: issara.c@psu.ac.th

**Abstract:** The demand for rubber gloves has significantly increased in both medical and non-medical fields due to the spread of the coronavirus in 2019. It is challenging for rubber glove manufacturing industries to balance the production and demand for the product. Additionally, they must determine techniques to decrease the production costs so as to make rubber gloves more economical for consumers. Generally, natural gas, fossil fuels, and renewable energy sources are used worldwide in the manufacturing of rubber gloves. In addition, Thailand uses biomass energy for rubber glove production, but biomass utilization is not economically friendly. This study used different biomasses as fuel in rubber glove production so as to reduce production costs and make the process more environmentally friendly. Wood chip (WC), palm kernel shells (PKS), and oil palm mesocarp fiber (OPMF) biomass were collected from local regions and used in different ratios. The samples of WC, PKS, and OPMF were prepared in four different ratios, namely, 88:12:0, 85:15:0, 85:13:2, and 85:10:5, for efficient biomass utilization. The 85:10:5 (WC: PKS: OPMF) ratio was found to be the optimal ratio as the annual production costs of rubber gloves significantly decreased to USD 1.64 per 1000 units of gloves. Furthermore, this biomass ratio also showed the best boiler efficiency of 74.87%. Therefore, WC, PKS, and OPMF biomass are recommended as fuel for rubber glove industries to make sustainable and economical production processes.

**Keywords:** biomass fuel; cost analysis; rubber glove; boiler efficiency; sustainable economy

## 1. Introduction

Owing to the COVID-19 pandemic, there has been an increased demand for single-use gloves, which has even exceeded the current production capacity of glove manufacturers by ≈215 billion units (or 37%) [1]. Although glove manufacturers have made their best efforts to increase production, industry experts have predicted a probable shortage of the product. However, there was a significant decrease in the demand for surgical gloves because more than 28 million surgeries were postponed or canceled during this pandemic [1]. However, after all the lockdown measures were lifted, surgeries increased, which accelerated the demand for these gloves. It was noted that North America dominated the medical glove market share globally, followed by Europe, APAC countries, Latin America, the Middle East, and Africa.

In the future, manufacturers will need to develop new products, collaborate, or expand into new territories to maintain revenue growth. For instance, Supermax, a major manufacturer, produces 24 billion units each year. The company plans to increase its manufacturing capacity to 44 billion pieces by 2024 [1]. It was noted that the COVID-19 pandemic significantly increased the demand for examination gloves by >82.32% as

compared to regular demand [2]. The major rubber glove manufacturing companies are concentrated in Southeast Asia, such as those in China, Indonesia, Japan, Malaysia, and Thailand. In this study, we focused on industries in Thailand. Thailand has 31 rubber glove manufacturers, where Sri Trang province produces more than 50% of the total output in the country. This has made Thailand the world's second biggest manufacturer of rubber gloves as it produces 20% of the global supply, making it a USD 28 billion industry. Out of the total production costs, the energy costs resulting from electricity, fuel, and water are 51%, 43%, and 6%, respectively. A total of 55% of the glove manufacturing industries use gas as their major raw material, while 45% use biomass. Most glove manufacturing industries use fuel to heat their steam boilers or thermal oil heaters (TOHs) to generate heat energy supplied to the production line. Furthermore, the vulcanizing process uses ≈ 80% of the total energy [3].

Boilers are usually used in the industries to generate steam needed for production. The safety and efficiency of the boilers directly affect their production rates. The use of alternative fuel sources and a decrease in the heat losses of the boilers could significantly decrease the costs of the boiler operations. Boiler process optimization is based on energy management [4]. Boilers are considered an important component in any industry or manufacturing unit as they are defined as a component where fuel is used to generate the necessary amount of heat energy needed for the manufacturing process. These are mainly used for converting water into steam at a constant pressure, which is then supplied to the production line at a constant pressure [4].

Many boiler systems operate at a ≤75% efficiency, irrespective of the different processes used to improve their efficiency. Researchers have highlighted the need to search for alternative renewable fuel sources for boilers as these can decrease the greenhouse emissions during boiler operations. The use of fossil fuels for producing electrical/heat energy in the industries has significantly increased greenhouse emissions, which has led to global warming [5]. Different boilers use various fuel forms based on their operating principles. Every fuel displays unique properties that affect combustion [5].

A boiler refers to a popular steam-generating system used in many industries and power plants. A significant percentage of the global energy consumed is used in boilers. Scheduled maintenance can improve boiler efficiency [6]. Figure 1 presents the glove manufacturing process, where raw biomass is used as the primary heat source. As shown in the figure, biomass is supplied to the TOH/steam boiler, which generates constant heat that is supplied to different locations in the production line, such as dipping lines, pre- and post-leaching tanks, cleaning tanks, polymer dipping tanks (polymer-coated process), coagulant dipping tanks, chlorine rinsing tanks (i.e., online chlorination), and ovens.

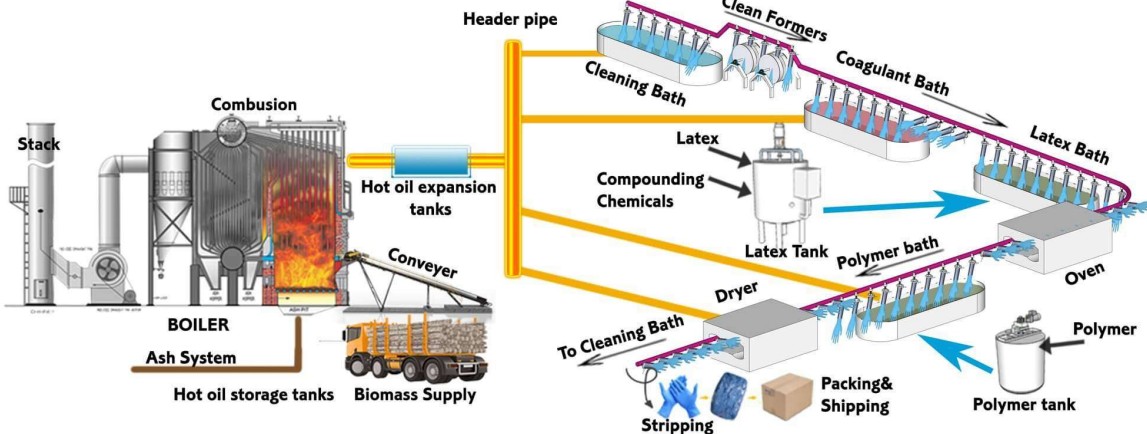

**Figure 1.** Process flow for biomass fuel feeding and heat supply in the glove manufacturing process.

Biomass is the third largest primary energy source after coal and oil [7]. In addition, biomass fuels are known to be extremely environmentally friendly. First, biomass combus-

tion does not produce $CO_2$ gas because biomass absorbs atmospheric $CO_2$ during growth for use in photosynthesis, which is equivalent to the amount released during combustion. Second, biomass burning also indirectly reduces the greenhouse gas $CH_4$ by preventing the release of $CH_4$ from biomass landfilling on agricultural land. In terms of global warming, $CH_4$ has a 21-times-stronger effect than $CO_2$ [8]. Third, most biomass contains very little or no sulfur. Therefore, the partial combustion of biomass from coal with a high sulfur content can reduce the $SO_2$ gas content. In addition, the combustion of high-sulfur coal and alkaline biomass (i.e., K and Na) in the biomass ash helps to capture some of the $SO_2$ produced during combustion [9]. Fourth, biomass fuels, such as wood and paper, contain less nitrogen than coal. In addition, biomass fuels release large amounts of $NH_3$ during the evaporation process [10]. This ammonia helps convert NO to $N_2$. Ammonia from biomass sources is considered a source of NOx reduction. Combining coal with biomass is a measure that can be used to reduce $NO_x$ from coal combustion. Finally, this helps to reduce soil and water pollution problems from landfills and biomass storage.

Total plant biomass consists of three main types of fibrous polymeric compounds: cellulose, hemicellulose, and lignin [11]. The proportions of the primary and secondary components of plants are variable. They depend on the species, type of plant tissue, age, and growing conditions. Biomass consists of the following elements: carbon (C), hydrogen (H), oxygen (O), nitrogen (N), sulfur (S), and chlorine (Cl). Ash from biomass contains the following important elements: Al, Ca, Fe, K, Mg, Na, P, Si, and Ti. The less important elements in the ash are AS, Be, Cd, Co, Cr, Cu, Hg, Mn, Mo, Ni, Pb, Sb, Tl, V, and Zn; of these elements in the ash, Ca and Mg lead to an increase in the ash's melting point, while K significantly lowers the ash's melting point. Chloride compounds and alkali silicate compounds present in the ash lower the ash's melting point temperature [12,13]. The weight percentages (dry) of C, H, and O in the biomass generally range from 30 to 60%, 5 to 6%, and 30 to 45%, respectively. N, S, and Cl are generally less than 1%. However, the nitrogen content is sometimes higher because it is an essential food for plant growth. The properties of biomass fuels, which differ from those of coal fuels, affect the behavior of the combustion reaction. For example, the formation of ash particles on the surface of steam tubes and the emissions of biomass are different from those of coal [14]. The differences in the properties of the two fuels can be summarized as follows:

- Compared to coal fuels, biomass fuels generally contain more volatile components and oxygen. They have a low carbon content and heating value.
- The pyrolysis process of biomass fuels starts at a lower temperature.
- The heat content generated from the vaporization of biomass is about 70% as compared to about 30% for coal.
- Biomass fuels such as rice straw and oil palm empty fruit branch contain more free alkali (K and Na, but mainly K) in the ash. This leads to more severe problems with ash melting, slagging, ash deposition on the heat exchanger surface, or fouling than coal.
- Biomass charcoal has a better oxidation response than coal charcoal because it has a larger surface area and alkali is present in the charcoal as a catalyst (Blasi et al., 1999).

The behavior of biomass, which differs from coal, affects the use of biomass for thermal utilization and the selection of appropriate biomass combustion technology.

Several other factors may be even more critical when using biomass fuels for thermal power generation, including a sustainable supply of biomass. Some types of biomasses may only be available for a few weeks each year. Therefore, biomass must be accumulated for use throughout the year. This is different from fossil fuels. Some types of biomass must be prepared before transportation to the combustor for further heat production, e.g., leaching, drying, or pelletizing.

Another drawback is the lack of technical data regarding how the fuel is transported into the combustion system, as well as technical data regarding the combustion characteristics and emissions of biomass fuels. In addition to the above obstacles, the private sector in Thailand still lacks appropriate and concrete incentive measures from the government. In addition, the nature of much potential biomass requires proper management across

several stages, from field collection to transportation and fuel processing. Until the fuel is transported to the boiler combustion chamber, the private sector, especially when investing in biomass fuel, lacks fuel (e.g., rice mills with chaff or sugar mills that already have bagasse as a by-product) and must source the fuel from outside sources. All of these factors have caused the cost per unit of production to increase to the point where the investment may no longer be worthwhile. Therefore, in addition to rice husks, bagasse, palm fruit, and palm kernel shells, mills in Thailand rarely use biomass. Instead, bark (from the paper industry) is a by-product of the respective industry.

Biomass is the most popular alternative energy source because it is a clean, inexpensive, and widely available renewable resource. There are various biomasses, such as energy crops, wood, agricultural residues, municipal wastes, and industrial wastes, and there are various ways by which biomass can be converted into energy. Incidentally, biomass is the only reliable resource that can be converted into all forms of energy, and the compression of biomass is the most important technique by which to obtain better properties in pellet form. In recent years, several studies have dealt with biomass pellets, such as wood waste and wood pellets, pellets from municipal solid waste, pellets from agricultural waste, pellets from sewage sludge, and pellets from industrial waste [15].

The palm oil industry produces significant amounts of solid waste. The solid wastes from plantations are oil palm trunk (OPT) and oil palm frond (OPF), while processing plants generate empty fruit bunches (EFB), OPMF, and PKS [16]. Table 1 summarizes the contents of lignocellulose in oil palm biomass and the proportions of cellulose, hemicellulose, lignin, and ash in oil palm biomass. Therefore, a proximate and ultimate analysis and the heat value of raw biomass fuel is show in Table 2.

**Table 1.** Composition of oil palm biomass.

| Biomass Type | Cellulose | Hemicellulose | Lignin | Ash | Reference |
|---|---|---|---|---|---|
| PKS | 28.8–27.2 | 21.6–22.7 | 44.0–50.7 | 8.6–16.3 | [16] |
| | 27.70 | 21.60 | 44.0 | No data | [15] |
| EFB | 34.0–40.4 | 17.2–22.4 | 23.1–29.6 | 5.0–6.5 | [16] |
| | 23.70 | 21.60 | 29.20 | No data | [15] |
| | 26.0 | 43.0 | 24.0 | No data | [15] |
| OPMF | 23.0–28.8 | 25.3–30.5 | 25.5–28.97 | 2.6–5.8 | [16] |
| | 19.0 | 37.0 | 33.0 | No data | [17] |
| OPF | 31.0–42.8 | 12.5–22.5 | 15.2–25.0 | 5.0–5.8 | [16] |
| OPT | 40.3–50.78 | 18.7–30.36 | 17.9–26.8 | 2.4–2.9 | [16] |

**Table 2.** Proximate and ultimate analysis and heat value of raw biomass fuel.

| Biomass | Proximate Analysis (wt, %) | | | | Ultimate Analysis (wt, %) | | | | | HHV | LHV | References |
|---|---|---|---|---|---|---|---|---|---|---|---|---|
| | M | FC | VM | Ash | C | H | O | N | S | (MJ/kg) | (MJ/kg) | |
| PKS | 10.00 | 23.00 | 74.00 | 3.00 | 45.10 | 50.10 | 49.20 | 0.56 | 0.04 | 17.58 | - | [18] |
| | 1.74 | 10.66 | 83.38 | 4.22 | 46.53 | 5.85 | 42.32 | 0.89 | 0.12 | 18.81 | - | [19] |
| | 11.00 | 19.70 | 67.20 | 2.10 | 49.74 | 5.32 | 44.86 | 0.08 | 0.16 | 16.30 | - | [15,16] |
| | 5.40 | 18.80 | 71.10 | 4.70 | 48.06 | 6.38 | 34.10 | 1.27 | 0.09 | - | - | [15] |
| | 10.23 | 1.42 | 85.11 | 3.24 | 47.88 | 5.15 | 42.69 | 0.94 | 0.10 | - | - | [20] |
| EFB | 66.00–69.00 | 10.80–14.50 | 86.50–87.70 | 3.70–5.30 | 48.72 | 7.86 | 48.18 | 0.25 | - | 18.88 | | [21] |
| | 8.78 | 8.60 | 79.65 | 3.00 | 48.79 | 7.33 | 40.18 | n.d. | 0.68 | 16.80 | - | [15] |
| | 54.10–56.50 | 8.0–8.2 | 34.3–34.7 | 2.04–2.16 | 21.00–22.80 | 2.70–2.90 | 16.70–18.30 | 0.41–0.42 | n.d. | 8.90–9.45 | 6.48–7.48 | [22] |
| | n/a | 27.90 | 67.50 | 4.60 | 40.70 | 5.40 | 47.80 | 0.30 | 1.20 | | | [23] |
| | 15.01 | 0.98 | 79.58 | 4.48 | 43.89 | 5.33 | 54.32 | 0.52 | 0.10 | - | - | [20] |
| OPF | 62.00–77.00 | 3.20–14.80 | 83.60–88.30 | 3.20–3.80 | 48.43 | 10.48 | 46.50 | 12.40 | - | 15.72 | | [21] |
| OPT | 67.00–81.00 | 4.90–7.80 | 68.30–88.30 | 2.90–3.70 | 51.41 | 11.82 | 51.16 | 0.17 | - | 17.47 | | [21] |
| OPMF | 30.40–33.40 | 14.10–14.70 | 51.10–51.70 | 2.24–2.36 | 33.10–36.10 | 3.40–3.80 | 25.20–27.60 | 1.16–1.20 | 0.09 | 13.87–17.87 | 11.57–13.37 | [22] |
| | n/a | 16.13 | 73.03 | 10.83 | 51.52 | 5.45 | 40.91 | 1.89 | 0.23 | 19.00 | | [15] |
| | 11.10 | 1.01 | 80.08 | 7.90 | 42.20 | 5.21 | 42.34 | 2.21 | 0.14 | - | - | [20] |
| WC | 8.50 | 14.75 | 83.09 | 0.83 | 46.39 | 5.75 | 14.71 | 0.02 | 0.00 | | | [23] |
| | 6.40 | 3.90 | 15.30 | 74.5 | 19.30 | 4.60 | 22.40 | 0.10 | 0.10 | 9.13 | | [24] |

In recent years, there has been a significant decrease in the use of fossil fuels, owing to the increased awareness of greenhouse gas emissions. Renewable sources or biomass fuel are effective alternatives in many industries [25]. This has led to massive growth in the consumption of renewable energy sources, such as traded renewable electricity sources (except biofuels and hydropower), with an increase of 3.2 exajoules, which was the highest increase in energy consumption volume since the data were first collected. In 2021, renewable energy sources accounted for 40% of global primary energy sources, higher than other forms of energy. However, in 2018, the contribution of renewable energy sources to the total energy increased from 4.5 to 5.0% [26].

After conducting a thorough literature review, we did not find any consolidated research on the utilization of biomass as fuel in glove manufacturing industries regarding improvements to energy usage, savings, related bill savings, offer and cost–benefit analyses, and its impact on greenhouse gas emissions. Here, we aimed to make the rubber glove manufacturing process more eco-friendly and more cost efficient. An optimum mixture of wood chip (WC, 80% *w/w*) and palm oil mill fiber (20% *w/w*) was proposed as a boiler fuel to reduce the per unit glove manufacturing cost. We fed this multi-biomass fuel to the TOH, supplying heat to the dipping lines. We used the existing formula for the TOH unit and air velocity. The calorific value, moisture content, biomass fuel consumption, and the production cost for 1000 gloves (USD/CTN) using this process were also analyzed.

## 2. Materials and Methods

### 2.1. Sample Preparation

The raw biomass fuel used in this study included residual agriculture material, such as WC, PKS, and OPMF. Compared to other deciduous trees, the WC material was acquired from para rubberwood as this plant is widely produced in South Thailand. Figure 2 presents the characteristics of each biomass fuel. Most of the biomass fuel used in this study was grown in or acquired from South Thailand.

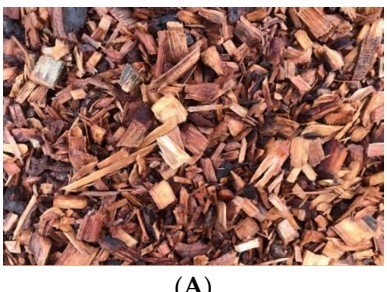 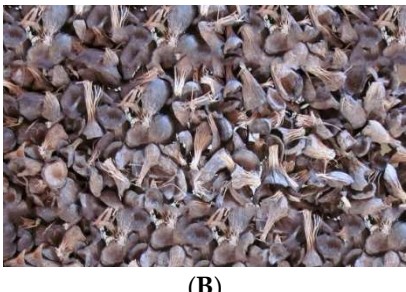 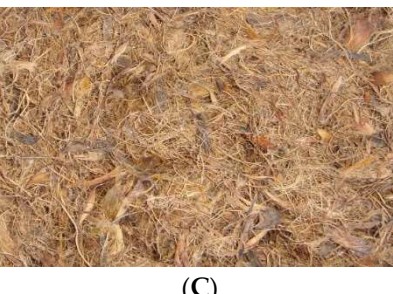

(**A**)  (**B**)  (**C**)

**Figure 2.** Characterization of the biomass fuels, such as wood chip and oil palm, used in this study, where (**A**) wood chip, (**B**) palm kernel shell (PKS), and (**C**) oil palm mesocarp fibers (OPMF).

The proximate analysis and moisture content of the biomass fuel were determined using the gravimetric method, with different factors such as volatile matter, HHV, fixed carbon, and ash content, based on the ASTM D7582 process. The hydrogen (H), carbon (C), nitrogen (N), oxygen (O), and sulphur (S) contents were also obtained using a CHNS/O analyzer (Flash 2000, Thermo Scientific, Milan, Italy) at the Office of Scientific Instrument and Testing (OSIT), Prince of Songkhla University, Thailand [27]. All the biomass fuel experiments were conducted using a TOH (10 mil. Kcal/h; type YGW-12000T, serial no. 2017A257) at a rated capacity of 12,000 KW, a rate outlet/inlet temperature of 320/285 °C, a design pressure of 0.8 Mpa, and a hydrostatic test pressure of 1.2 MPa, built-in January 2017 (Maxtwo Engineering and Services Sdn., Hhd) at the rubber glove manufacturing industry in Songkhla Province, Thailand.

### 2.2. Experiment Set Up

The biomass was stored so that the moisture was below 50%. Each biomass was weighed before mixing the recipe, and then the biomass loader was moved back and forth for at least 15 min to ensure that the mixture was homogeneous. The conveyor brought mixed biomass fuel (MBA, MBB, MBC, and MBD) to the TOH fuel chamber. The mixed biomass fuel was burned there, and the combustion heat led to the oil heating system. This hot oil was fed to each area of the glove production line (cleaning tanks, coagulant dip tanks, latex dip tanks, pre- and post-leaching tanks, and all ovens). The single and mixed biomass fuels samples were collected for the proximate and ultimate analysis. Fuel consumption, heat consumption, and glove output were monitored, and each recipe was continued for three months. The wastes generated during combustion were ash and pollutant emissions. The ash was disposed of in a landfill. The use of biomass ash has been investigated in many studies, such as road construction, cement industry, fertilizers, and bricks. Pollutant emissions were analyzed for this study. The emitted air pollution was collected from the stack of TOH. The process flow for biomass fuel mixing, feeding, and heating supply to glove process is illustrated in Figure 3.

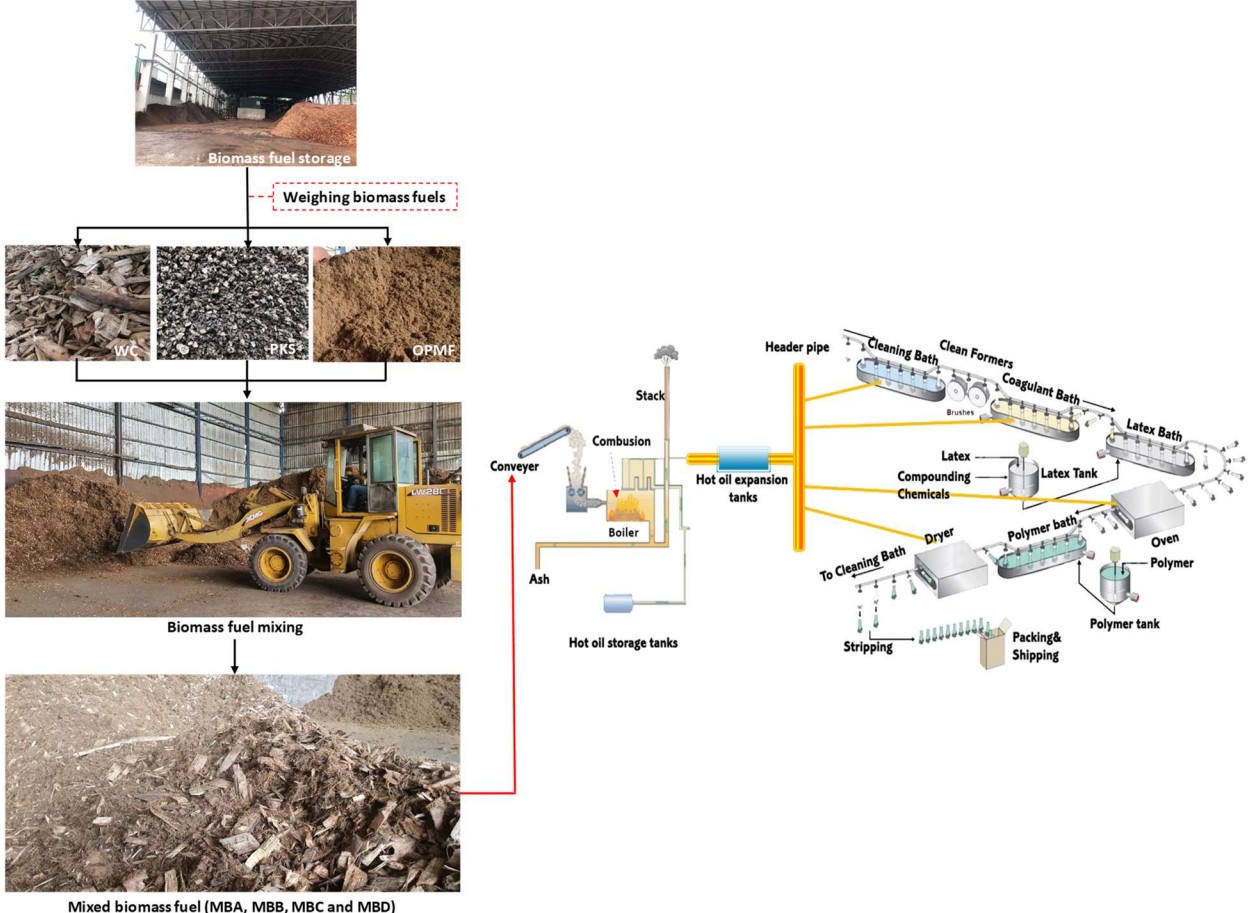

**Figure 3.** Biomass fuel mixing, feeding, and heating supply to glove process.

### 2.3. Biomass Fuel Analytical Methods

Moisture content of biomass fuel: Gravimetric method/ASTM D 3173.
Proximate analysis: ASTM D 5468.
Ultimate analysis
Moisture content: ASTM D 7582.
Volatile matter: ASTM D 7582.
Fixed carbon: ASTM D 7582.
Ash value: ASTM D 3174.

Gross heating value (as received basis): Bomb Calorimeter/AC-350/BSEN 14918.

Heat energy consumption: Read data from thermal hot oil machine and one-way ANOVA analysis.

Moisture content of biomass fuel: Gravimetric method/ASTM D 3173.

Heat energy consumption: Read data from thermal hot oil machine and one-way ANOVA analysis.

Production cost: One-way ANOVA analysis

Total suspension particulate: US.EPA Method 5.

Sulfur dioxide: US.EPA Method 6C.

Oxide of nitrogen: US.EPA Method 7E.

Carbon monoxide: US.EPA Method 10.

Carbon dioxide: US.EPA Method 3.

### 2.4. Experimentation and Observation of Boiler Performance Parameters

Four different mixing ratios of the biomass materials, i.e., WC: PKS: OPMF in the ratios of 88:12:0, 85:15:0, 85:13:2, and 85:10:5 *w/w* (Table 3) were investigated. Manufacturing industries could use these different ratios to obtain good-quality biomass material at low prices, allowing them to decrease production costs. WC was observed to be cheaper than PKS but higher than OPMF, while OPMF and PKS had a lower heating value than WC. Hence, a higher concentration of WC than PKS and OPMF was considered in the four different combinations. A proximate analysis was carried out, and we determined the HHV of the raw biomass. Furthermore, the moisture content of the raw biomass was estimated before mixing.

**Table 3.** Ratio of the different raw biomass materials.

| Sample | WC | PKS | OPMF |
|---|---|---|---|
| Mixed biomass sample A (MBA) | 88 | 12 | - |
| Mixed biomass sample B (MBB) | 85 | 15 | - |
| Mixed biomass sample C (MBC) | 85 | 13 | 2 |
| Mixed biomass sample D (MBD) | 85 | 10 | 5 |

### 2.5. Performance Analysis of Boiler

The performance parameters of boilers and thermal oil boilers decrease over time for many reasons, such as poor heat transfer, improper combustion, poor operation, poor maintenance, and bad fuel quality. Hence, a proper balance between mass (amount of combustion air) and energy (heat loss) is needed to determine boiler efficiency. An energy balance assists in assessing the boiler efficiency as it determines the unavoidable and avoidable heat losses. Even the combustion air affects the boiler efficiency.

Thermal efficiency is the measure of the selected instruments and operations used to transfer combustion heat into steam or oil. Thus, boiler efficiency refers to the "percentage of usable input heat used for steam generation". Two different techniques can be used to assess boiler efficiency, i.e., direct or indirect. The direct technique determines the boiler efficiency as a ratio of useful heat output generated by the boiler and heat input. Accurate calculation of useful heat generated by the boiler and HHV of the fired fuel is needed to determine the boiler efficiency. However, the indirect technique is called the heat loss technique. It presents the accurate effect of the individual heat losses on the boiler efficiency.

Thermal oil efficiency is calculated using Equation (1):

$$\eta = \frac{Q_{out}}{Q_{in}} \times 100 \tag{1}$$

where

$\eta$: Boiler efficiency (%).

$Q_{out}$: Energy output (kJ/h), refer to Equation (2).

$Q_{in}$: Total energy input (kJ/h), refer to Equation (4).

$$Q_{out} = m \times Cp \times \Delta T \tag{2}$$

where

$m$: Flow rate (kg/h).
$Cp$: Specific heat (kJ/kg °C).
$\Delta T$: Temperature difference (°C), refer to Equation (3).

$$\Delta T = T_{out} - T_{in} \tag{3}$$

where

$T_{in}$: Inlet temperature (°C).
$T_{out}$: Outlet temperature (°C).

$$Q_{in} = Q_{inWC} + Q_{inPKS} + Q_{inOPMF} \tag{4}$$

where

$Q_{in}$: Total energy input (kJ/h).
$Q_{inWC}$: Energy input of WC (kJ/h).
$Q_{inPKS}$: Energy input of PKS (kJ/h).
$Q_{inOPMF}$: Energy input of OPMF (kJ/h).

*2.6. Economics of Biomass Utilization*

Aside from environmental sustainability, biomass fuels must also be economically competitive to attract investors while significantly contributing to changing the overall balance of primary energy consumption [28].

In this section, three sets of costs are presented to economically evaluate fuel, operation, and maintenance costs. To determine the net present value (NPV), the present value of the estimated cash flows was calculated based on a previously determined rate of return. The internal rate of return (IRR) was calculated as the interest rate that sets the NPV to zero. A payback period based on the discounted cash flows is also presented. Equations used for the economic analysis are presented in Equations (5)–(7) [29]:

$$NPV = \sum_{t=0}^{n} (C_b - C_C)_t (1 + i))^{-t} \tag{5}$$

$$PBP = \sum_{t=1}^{Pt} (C_b - C_c)_t (1 + i)^{-t} = 0 \tag{6}$$

$$IRR = \sum_{t=0}^{n} (C_b - C_c)_t (1 + irr)^{-t} = 0 \tag{7}$$

where $C_b$ is the cash benefit of the investment; $C_c$ is the cash cost of investment; $(C_b - C_c)_t$ is the net cash flow in the year ($t$); $n$ is the calculation period, which is equal to the project lifecycle; and $i$ is the cut-off discount rate [29].

*2.7. Cost Analysis Calculation*

Cost analysis: WC: PKS: OPMF in the ratio of 85:13:2 % *w/w*.

Actual glove production: 5,272,796 pcs/day (example for calculation). The fuel cost per carton can calculate as per Table 4.

**Remark 1.** *1 United States Dollar equals 35.43 Thai Baht*

Glove production  = 5,277,796.00   pcs/day;
         = 5,277,796.00/1000;

|  |  |  |
|---|---|---|
| | = 5277.80 | carton/day. |
| Fuel cost | = 327,812.30 | Thai baht (THB) (refer to the table above). |
| Cost per carton | = 327,812.30/5277.80; | |
| | = 62.11 | THB/Carton. |
| 1 United States Dollar equals 35.43 | | Thai Baht; |
| | = 62.11/35.73; | |
| | = 1.75 | USD/carton or USD/1000 gloves. |

**Table 4.** Fuel cost per carton (1000 pieces of glove) calculation.

| Biomass Type | Fuel Consumption (kg/day) | Ration (% *w/w*) | Price (THB/KG) | Cost (THB) |
|:---:|:---:|:---:|:---:|:---:|
| WC | 213,890.00 | 85 | 1.09 | 233,140.10 |
| PKS | 31,890.00 | 13 | 2.83 | 90,248.70 |
| OPMF | 4915.00 | 2 | 0.90 | 4423.50 |
| Total | 250,695.00 | 100 | | 327,812.30 |
| Cost per carton (THB/Carton) | | | | 62.11 |
| Cost per carton (THB/Carton) | | | | 35.43 |

*2.8. Environmental Impact*

Environmental assessments can be accomplished through different approaches and technologies. The focus of this research was to evaluate and mitigate the environmental hazards associated with rubber glove manufacture utilizing biomass fuel. Many risk factors have been investigated as a result of environmental concerns. The environmental study was carried out in accordance with the seven standards of the US Environmental Protection Agency (USEPA), Thailand's Pollution Control Department (PCD), and Thailand's Department of Industrial Work (WIP). Environmental standards were compared to measured environmental data from the environmental monitoring systems (Apex Instruments, XC-572) on the site. We used the notification of the Ministry of Industry on the prescription of the content value of air contaminants emitted from the factory B.E. 2549 [1]. The reference condition was 25 degrees Celsius at 1 atmosphere, with excess $O_2$ of 7.0% and dry conditions [1].

**3. Results**

Table 5 presents the proximate and ultimate analysis of the raw biomass fuel used in this study. The OPMF biomass fuel showed the highest average moisture content of 42.71% *w/w*, followed by PKS (11.20% *w/w*), while WC showed the lowest moisture content value of 9.76% *w/w*. PKS showed the highest gross calorific value (GCV), or high heating value (HHV), of 17,266 kJ/h, whereas WC and OPMF showed values of 16,491 and 11,096 kJ/h, respectively. Regarding the net calorific value (NCV), or low heating value (LHV), PKS showed the highest value of 16,054 kcal/h, whereas both WC and OPMF showed lower values of 15,192 and 9783 kJ/h, respectively. Both the proximate and ultimate analysis results were consistent with those of other research studies (see Table 2).

The current study found that moisture did not affect the calorific value or heat consumption, but the combustion of the mixed-fiber energy showed a higher value. Interestingly, there was no significant change in the temperature value or energy consumption in all four biomass mixes compared to the usage/non-usage of fiber in the biomass fuel mixture regarding HHV (Figure 3).

Combustion of the mixed-fiber fuel showed a higher value of 50.38%. However, none of the four biomass fuel combinations showed a significantly different temperature value. Additionally, the energy consumption was not significantly different as compared to the usage/non-usage of fiber in the biomass fuel mixture regarding HHV (Figure 4).

**Table 5.** Proximate and ultimate analysis result of biomass fuels.

| No. | Parameters | Units | WC | PKS | OPMF | MBA | MBB | MBC | MBD |
|---|---|---|---|---|---|---|---|---|---|
| 1 | Moisture | % wt. | 9.76 ± 0.02 | 11.20 ± 0.02 | 42.71 ± 0.30 | 10.02 ± 0.05 | 9.97 ± 0.11 | 10.67 ± 0.01 | 10.28 ± 0.05 |
| 2 | Volatile matter | % wt. | 74.49 ± 0.44 | 66.80 ± 0.14 | 44.28 ± 0.13 | 73.21 ± 0.18 | 73.63 ± 0.26 | 73.30 ± 0.47 | 73.91 ± 0.13 |
| 3 | Fixed carbon | % wt. | 14.67 ± 0.44 | 18.93 ± 0.15 | 10.10 ± 0.17 | 15.98 ± 0.22 | 15.43 ± 0.24 | 15.04 ± 0.47 | 15.02 ± 0.18 |
| 4 | Ash content | % wt. | 1.08 ± 0.02 | 3.08 ± 0.06 | 2.90 ± 0.01 | 0.79 ± 0.02 | 0.97 ± 0.03 | 0.99 ± 0.02 | 0.78 ± 0.01 |
| 5 | HHV | kJ/kg | 16,850 ± 97 | 17,154 ± 376 | 11,196 ± 281 | 16,737 ± 53 | 16,422 ± 81 | 17,110 ± 57 | 17,249 ± 44 |
| 6 | Carbon (C) | % wt. | 44.74 ± 0.18 | 45.93 ± 0.12 | 30.90 ± 0.89 | 46.79 ± 0.25 | 44.55 ± 0.12 | 45.31 ± 0.11 | 46.23 ± 0.15 |
| 7 | Hydrogen (H) | % wt. | 6.05 ± 0.16 | 5.64 ± 0.01 | 6.11 ± 0.11 | 6.01 ± 0.07 | 6.01 ± 0.04 | 6.02 ± 0.21 | 5.96 ± 0.14 |
| 8 | Nitrogen (N) | % wt. | 0.28 ± 0.01 | 0.41 ± 0.01 | 0.76 ± 0.03 | 0.25 ± 0.01 | 0.49 ± 0.01 | 0.28 ± 0.02 | 0.31 ± 0.01 |
| 9 | Sulphur (S) | % wt. | <0.01 | <0.01 | <0.01 | <0.01 | <0.01 | <0.01 | <0.01 |
| 10 | Oxygen (O) | % wt. | 42.09 ± 0.70 | 36.80 ± 0.50 | 46.22 ± 0.36 | 42.73 ± 0.53 | 42.54 ± 0.54 | 41.73 ± 0.62 | 41.56 ± 0.22 |
| 11 | O: C ratio | - | 0.94 | 0.80 | 1.50 | 0.91 | 0.96 | 0.92 | 0.90 |
| 12 | H: C ratio | - | 0.14 | 0.12 | 0.20 | 0.13 | 0.14 | 0.13 | 0.13 |
| 13 | GHV | kJ/kg | 16,491 ± 173 | 17,266 ± 53 | 11,096 ± 342 | 17,022 ± 100 | 16,296 ± 9 | 16,717 ± 332 | 16,975 ± 153 |
| 14 | NHV | kJ/kg | 15,192 ± 139 | 16,054 ± 52 | 9783 ± 333 | 15,732 ± 92 | 15,004 ± 1 | 15,423 ± 287 | 15,694 ± 124 |

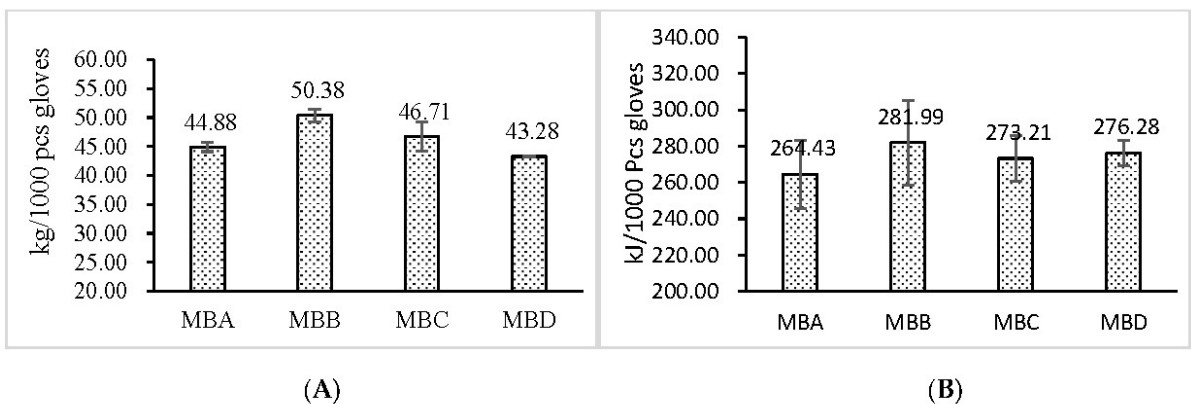

**Figure 4.** (**A**) Fuel consumption and (**B**) heat consumption of each biomass fuel mixture.

The MBD formulation showed a lower fuel consumption (43.28 kg/1000 units of gloves) than that of the MBA, MBB, and MBC formulations, i.e., 44.88, 50.38, and 46.71 kg/1000 units of gloves, respectively. However, MBA showed a lower heat consumption (264.43 kJ/1000 units of gloves) as compared to MBB, MBC, and MBD, i.e., 281.99, 273.21, and 276.28 kJ/1000 units of gloves, respectively.

A decrease in the production cost was noticed when OPMF was added to the biomass fuel mixture. The production costs for 1000 units of gloves were USD 2.06, 2.10, 1.85, and 1.62 for the MBA, MBB, MBC, and MBD formulations, respectively. Furthermore, if the glove manufacturing industries used the MBD mixture, the annual costs could decrease by USD 32.37 k (a decrease from 287.94 to 255.57 k USD/Year) as shown in Figure 5. Figure 5 presents the trend of production cost/carton when the glove manufacturers did not buy fiber or improve the boiler efficiency since it increased the production costs.

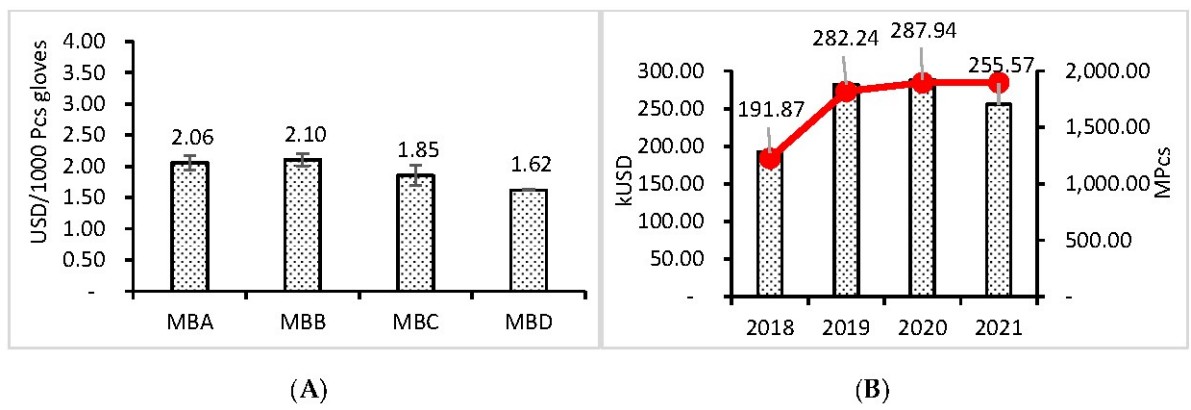

**Figure 5.** (**A**) Comparison of production costs and (**B**) predicted annual costs for 2021 with the MBD mixture.

The performance of TOH when using the direct method [30] did not show any significant differences for the mixtures. All the results ranged from 70.57 to 74.87% (Figure 6). Biomass combustion estimation is based on different steps, i.e., heating up, drying, devolatilization for generating char and volatiles, wherein volatiles include gases and tars, and combustion of volatiles and char [31].

$$\text{Wet biomass} \rightarrow \text{heating up/drying} \rightarrow \text{dry biomass} \tag{8}$$

$$\text{Biomass} \rightarrow \text{volatiles (tar and gases)} \rightarrow \text{char} \tag{9}$$

$$\text{Volatiles + air} \rightarrow \text{CO + CO}_2 \text{ (+PAH + unburnt hydrocarbons + soot + inorganic aerosols)} \tag{10}$$

$$\text{Char + air} \rightarrow \text{CO + CO}_2 \tag{11}$$

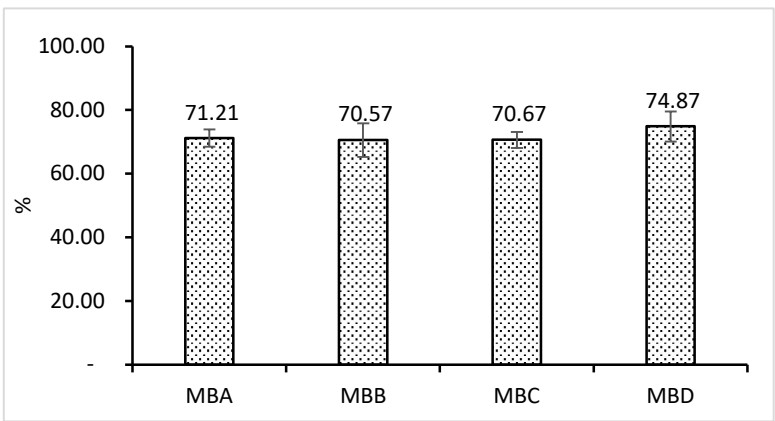

**Figure 6.** Efficiency of thermal oil boiler.

This study estimated the NPV, IRR, and PBP for multi-biomass fuels mixes. The NPV, IRR, and PBP were USD 51.81 K, 213%, and 10.67 months, respectively. The project was considered an attractive investment opportunity as the NPV was more than zero and the IRR was higher than 12% [32].

It was determined that the quality of air emissions from the biomass TOH case study met the requirements of Thailand's Department of Pollution Control (Ministry of National Research and Environment), Thailand's Department of Industrial Works (Ministry of Industry), and the United States Environmental Protection Agency (EPA). At two local towns within a five-kilometer radius of the rubber gloves plant, the following four key air quality factors were studied: $SO_2$, $NO_2$, CO, and total suspended particulate (TSP). The average values of $SO_2$ (one-hour average) and $NO_2$ (one-hour average) are shown in Table 6.

**Table 6.** Quality of air emission from the biomass TOH.

| Description | Unit | Method of Analysis | Results | Standard | Evaluation |
|---|---|---|---|---|---|
| Stack height | m. | Measuring tape | 30.00 | - | - |
| Diameter | cm. | Measuring tape | 120.00 | - | - |
| Cross-sectional area | $m^2$ | Measuring tape | 1.13 | - | - |
| Atmospheric pressure | mm.Hg | U.S.EPA Method 2 | 749.00 | - | - |
| Temperature | °C | U.S.EPA Method 2 | 199.17 | - | - |
| Gas velocity | m/s | U.S.EPA Method 2 | 21.74 | - | - |
| Flow rate | $m^3$/h | U.S.EPA Method 2 | 88,530.68 | - | - |
| Oxygen ($\%O_2$) | % | U.S.EPA Method 3 | 10.50 | - | - |
| Carbon dioxide ($\%CO_2$) | % | U.S.EPA Method 3 | 5.40 | - | - |
| Nitrogen and other ($\%N_2$) | % | U.S.EPA Method 3 | 84.10 | - | - |
| Moisture content | % | U.S.EPA Method 4 | 7.67 | - | - |
| Isokinetic | % | U.S.EPA Method 5 | 99.09 | - | - |

**Table 6.** *Cont.*

| Description | Unit | Method of Analysis | Results | Standard | Evaluation |
|---|---|---|---|---|---|
| Total suspended particulate [2] | mg/m$^3$ | U.S.EPA Method 5 | 266.19 | ≤320 [1] | Pass |
| Sulfur dioxide [2] | ppm | U.S.EPA Method 6C | 1.34 | ≤60 [1] | Pass |
| Oxide of nitrogen [2] | ppm | U.S.EPA Method 7E | 66.83 | ≤200 [1] | Pass |
| Carbon monoxide [2] | ppm | U.S.EPA Method 10 | 634.86 | ≤690 [1] | Pass |

Remark: [1] Notification of Ministry of Industry on the prescription of the content value of air contaminants emitted from the factory B.E. 2549. [2] Reference condition was 25 °C at atmospheric pressure, excess $O_2$ of 7.0%, and dry conditions.

## 4. Discussion

Table 2 presents the results of the proximate analysis of the raw biomass (i.e., WC, PKS, and OPMF) and the mixed biomass fuel (i.e., MBA, MBB, MBC, and MBD). The moisture content in the fuel material significantly affects its energy value and combustion performance. The moisture content of biomass varies greatly depending on the type of biomass and the separation process. High moisture in biomass results in low adiabatic temperatures and causes the fuel to remain in the combustion chamber longer. Previous studies have stated that a good fuel material should contain a moisture content of ≤8–12% *w/w*. This type of fuel can show reasonable and sustainable combustion. Furthermore, the moisture content of the raw biomass in this study was found to range between 14.8 and 41.8% *w/w*. This indicated that they contained a high moisture content, which could affect their energy value and combustion performance. The OPMF material had the highest moisture content percentage compared to the feedstock-based biomass (ranging between 10 and 15% *w/w*) and was considered unsuitable. Thus, the fuel pellets were an efficient and sustainable option as compared to the raw biomass residue and could be used for firing boilers in glove manufacturing industries.

Additionally, the concentration of the fixed carbon and other volatile combustible compounds is directly responsible for the fuel's calorific value. The raw biomass contained a higher volatile content, indicating that it was easily combustible and ignitable, as compared to the mixed biomass with a low volatile content (Table 2). Solid carbon compounds are the main heat-generating sources during combustion. The mixed fuels contained a higher concentration of solid carbon compounds than the raw biomass material, significantly increasing their heating value. Thus, the mixed biomass fuel was more sustainable and could be effectively used for power generation.

Furthermore, the results presented in Table 2 indicate that the mixed biomass fuels contained a low ash content compared to the raw biomass. This could be because the mixed biomass fuel was solid and burned sustainably and slowly as compared to the raw biomass that burned incompletely and more rapidly, generating ash. In boilers, a high ash content is undesirable as it leads to the slagging of water tubes. It also decreases the combustion and conveying capacity, reduces boiler and combustion efficiency, and increases conveyor costs. The results indicated that the mixed biomass fuels showed a lower ash content than the raw biomass. Thus, it is concluded that the mixed biomass fuels (i.e., MBA, MBB, MBC, and MBD) are better fuels for firing TOHs in glove manufacturing industries.

## 5. Conclusions

In this study, we carried out a proximate and ultimate analysis of different raw biomass and mixed biomass fuels derived from WC, PKS, and OPMF. After comparing the different types of biomass and mixed biomass fuels, we noted that the moisture and ash contents of MBD was significantly lower compared to those of MBA, MBB, and MBC. MBD was found to have the lowest sulphur content. All the results indicated that the mixed biomass fuels were better suited for combustion in boilers than the raw biomass. Furthermore, the mixed biomass fuels used to fire the boilers decreased the boiler's maintenance costs. After testing the biomass materials, we noted that the mixed fuel with an optimal ratio of

85:10:5 (WC/PKS/OPMF) generated the lowest production costs of USD 1.64 per 1000 units of gloves.

Additional studies need to be conducted after collaborating with boiler manufacturers to improve the biomass fuel combustion capacity using torrefaction technology to decrease the moisture content. However, manufacturers need to assess their investment balance before making any decisions. In the future, we will thoroughly study the relationship between temperature and heat input in production line control.

**Author Contributions:** Conceptualization, I.C. and L.T.-O.; methodology, L.T.-O.; writing—original draft preparation, writing—review and editing, L.T.-O., S.C. and A.K.; supervision, K.-A.T. All authors have read and agreed to the published version of the manuscript.

**Funding:** This research received no external funding.

**Acknowledgments:** The authors are grateful to and acknowledge the Sustainable Innovation Center (SIC-PSU) and the Faculty of Environmental Management (FEM-PSU) for the technical equipment and facility support.

**Conflicts of Interest:** The authors declare no conflict of interest.

## Nomenclature

| | |
|---|---|
| WC | Wood chip |
| PKS | Palm kernel shells |
| OPMF | Oil palm mesocarp fiber |
| TOH | Thermal oil heater |
| CTN | Carton, or 1000 pieces of gloves |
| HHV | High heating value |
| LHV | Low heating value |
| GCV | Gross calorific value |
| NCE | Net calorific value |
| H | Hydrogen |
| C | Carbon |
| N | Nitrogen |
| O | Oxygen |
| S | Sulphur |
| MBA | Mixed biomass fuel sample A |
| MBB | Mixed biomass fuel sample B |
| MBC | Mixed biomass fuel sample C |
| MBD | Mixed biomass fuel sample D |
| η | Boiler efficiency (%) |
| $Q_{out}$ | Energy output (kJ/h), refer to Equation (2) |
| $Q_{in}$ | Total energy input (kJ/h), refer to Equation (4) |
| m | Flow rate (m$^3$/h) |
| Cp | Specific heat (kJ/kg °C) |
| ΔT | Delta temperature (°C), refer to Equation (3) |
| $T_{in}$ | Temperature inlet (°C) |
| $T_{out}$ | Temperature outlet (°C) |
| $Q_{in}$ | Total energy input (kJ/h) |
| $Q_{inWC}$ | Energy input of WC (kJ/h) |
| $Q_{inPKS}$ | Energy input of PKS (kJ/h) |
| $Q_{inOPMF}$ | Energy input of OPMF (kJ/h) |
| NPV | Net present value |
| PBP | Payback period |
| IRR | Internal rate of return |
| $C_b$ | The cash benefit of the investment |
| $C_C$ | The cash cost of invertment |
| $(C_b - C_C)_t$ | The net cash flow in the year (*t*) |

| n | The calculation period, which is equal to the project lifecycle |
|---|---|
| i | The cut-off discount rate |
| $SO_2$ | Sulphur dioxide |
| $NO_2$ | Nitrous oxide |
| CO | Carbon monoxide |
| TSP | Total suspended particulate |
| EPA | Environmental Protection Agency |

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
