# Peer review of "Optimization of Biomass Fuel Composition for Rubber Glove Manufacturing in Thailand"

_sustainability, doi:10.3390/su141912493_

Round 1
Reviewer 1 Report
Despite an interesting topic, especially in a time of global energy crisis, the title and parts of the discussion are misleading. Some acronyms are not defined/elaborated in the text, such as MBA, MBB, MBC, MBD etc. (MBA is not equal to MB-A). In the abstract author mentioned, “Generally, natural gas, fossil fuels and renewable energy sources are used worldwide in manufacturing rubber gloves”, what else do we have? “However, ASEAN countries, Vietnam, Malaysia, and Thailand still use the traditional techniques for rubber glove production” What are that traditional techniques? The authors have not studied any new method in this study for gloves manufacturing!
It is not clear how and why the authors selected those biomass ratios. There are some ordinary analyses and repetitions; for example, if you show fuel consumption, there is little point in showing heat consumption again in Fig. 3- It's very simple! How did the author calculate the dollar value of 1000 pcs gloves? The methodology for economic elevation is not self-explanatory. Also, as we know, the boiler's efficiency primarily depends on the CV of the fuel, and as there is almost no difference in the CV of fuel samples studied, the boiler efficiency is therefore nearly the same. Considering chemical composition and error, these biomass samples are not different and hence should not be considered a unique study. Overall, I don’t find anything significant from the study.
Author Response
- Despite an interesting topic, especially in a time of global energy crisis, the title and parts of the discussion are misleading. Some acronyms are not defined/elaborated in the text, such as MBA, MBB, MBC, MBD etc. (MBA is not equal to MB-A). In the abstract author mentioned, “Generally, natural gas, fossil fuels and renewable energy sources are used worldwide in manufacturing rubber gloves”, what else do we have? “However, ASEAN countries, Vietnam, Malaysia, and Thailand still use the traditional techniques for rubber glove production” What are that traditional techniques? The authors have not studied any new method in this study for gloves manufacturing!
Authors Reply:
Authors Would like to think to reviser for valuable comments. Authors added Acronyms in manuscript,
line 30-73.
There are many sources of energy such as natural gas, LPG, heavy oil, electricity, biomass fuel and diesel.
Authors removed ‘traditional techniques’ from abstract the main objective of this research to reduce the production cost focus on energy utilization.
Line 17-19
- It is not clear how and why the authors selected those biomass ratios. There are some ordinary analyses and repetitions; for example, if you show fuel consumption, there is little point in showing heat consumption again in Fig. 3- It's very simple! How did the author calculate the dollar value of 1000 pcs gloves? The methodology for economic elevation is not self-explanatory. Also, as we know, the boiler's efficiency primarily depends on the CV of the fuel, and as there is almost no difference in the CV of fuel samples studied, the boiler efficiency is therefore nearly the same. Considering chemical composition and error, these biomass samples are not different and hence should not be considered a unique study. Overall, I don’t find anything significant from the study.
Authors response
Authors would like to thanks to reviser for insightful comments. There are many biomasses ratio have been study on power station but it’s not available in boiler in robber glove production. From the literature revise authors best utilization in robber glove production. Production cast added in the manuscript. Cost analysis have been added.
Newly added
Actual glove production: 5,272,796Pcs/Day
Fuel using: WC: PKS: OPMF in ratio 85: 13: 2 %w/w
Biomass type |
Fuel consumption (kg/day) |
Ration (%W/W) |
Price (THB/KG) |
Cost (THB) |
WC |
213,890.00 |
85 |
1.09 |
233,140.10 |
PKS |
31,890.00 |
13 |
2.83 |
90,248.70 |
OPMF |
4,915.00 |
2 |
0.90 |
4,423.50 |
Total |
250,695.00 |
100 |
327,812.30 |
Glove production = 5,277,796.00 Pcs/day
= 5,277,796.00/1000
= 5,277.80 Carton/day
Fuel cost = 327,812.30 Thai baht (THB) (Refer above table)
Cost per carton = 327,812.30/5,277.80
= 62.11 THB/Carton
1 United States Dollar equals 35.43 Thai Baht
= 62.11/35.73
= 1.75 USD/Carton or USD/ 1,000 gloves

Reviewer 2 Report
The paper titled "Biomass Fuel Ratio Optimization for Rubber Glove Manufacturing Cost Reduction in Thailand" deals with an interesting topic. The following comments/concerns should be addressed before accepting the paper for publication.
Comments 1: Lines 31-33, provide a reference for the claim.
Comments 2: Line 40, “&” should be replaced with ‘and’. In professional writing w
Comments 3: Likewise, lines 42-44 need a reference for the claim.
Comments 4: Line 53, “the energy costs resulting from electricity, fuel and water are 51, 43 and 6%, respectively.” 52 and 41 what? Percentage? Please specify.
Comments 5: Too many abbreviations have been used in the manuscript. Thus, I highly recommend adding a nomenclature.
Comments 6: What is the rationale behind selecting the four conditions (88:12:0, 85:15:0, 85:13:2 and 85:10:5)? Why only four? Why not other values?
Comments 7: Lines 187-188, “OPMF biomass fuel showed the highest average moisture content of 42.71%, 187 followed by PKS (11.20%), while WC showed the lowest moisture content value of 9.76%.” what %? Vol. or wt.? Please specify clearly.
Comments 8: Likewise, lines 253-254, “moisture 252 content of ≤8-12%” “Furthermore, the moisture content of raw biomass ranged between 14.8 and 41.8%” what %? Vol. or wt.? Please specify clearly.
Comments 9: Same goes for line 257. What %? Vol. or wt.? Please specify clearly.
Comments 10: What’s the meaning of using full form and abbreviations together repeatedly? For the first term, it is correct but later authors should use only abbreviations. Otherwise, remove abbreviations. For example, lines 280-281, lines 106-107 and lines 25-26.
Comments 11: This article lacks references. Only 14 references were provided. Many of them do not use proper reference styles. Authors must cite relevant references in due places.
Author Response
The paper titled "Biomass Fuel Ratio Optimization for Rubber Glove Manufacturing Cost Reduction in Thailand" deals with an interesting topic. The following comments/concerns should be addressed before accepting the paper for publication.
Authors response: Authors would to thanks to reviser for valuable comments.
1.1 Comments 1: Lines 31-33, provide a reference for the claim.
Authors response: references added
1.2 Comments 2: Line 40, “&” should be replaced with ‘and’. In professional writing w
Authors response: Changed following reviewer suggestion, please see at line 84.
1.3 Comments 3: Likewise, lines 42-44 need a reference for the claim.
Authors response: Referenced added with highlighted in the manuscript
- Bapat, S. Kulkarni, and V. Bhandarkar, "Design and operating experience on fluidized bed boiler burning biomass fuels with high alkali ash," American Society of Mechanical Engineers, New York, NY (United States), 1997.
[9] M. Sami, K. Annamalai, and M. Wooldridge, "Co-firing of coal and biomass fuel blends," Progress in energy and combustion science, vol. 27, no. 2, pp. 171-214, 2001.
[10] K. Hein and J. Bemtgen, "EU clean coal technology—co-combustion of coal and biomass," Fuel processing technology, vol. 54, no. 1-3, pp. 159-169, 1998.
[11] P. Glarborg, A. Jensen, and J. E. Johnsson, "Fuel nitrogen conversion in solid fuel fired systems," Progress in energy and combustion science, vol. 29, no. 2, pp. 89-113, 2003.
[12] A. Williams, M. Pourkashanian, and J. Jones, "Combustion of pulverised coal and biomass," Progress in energy and combustion science, vol. 27, no. 6, pp. 587-610, 2001.
[13] T. Madhiyanon, P. Sathitruangsak, S. Sungworagarn, S. Pipatmanomai, and S. Tia, "A pilot-scale investigation of ash and deposition formation during oil-palm empty-fruit-bunch (EFB) combustion," Fuel processing technology, vol. 96, pp. 250-264, 2012.
[14] I. Obernberger, T. Brunner, and G. Bärnthaler, "Chemical properties of solid biofuels—significance and impact," Biomass and bioenergy, vol. 30, no. 11, pp. 973-982, 2006.
[15] C. Yin, L. A. Rosendahl, and S. K. Kær, "Grate-firing of biomass for heat and power production," Progress in Energy and combustion Science, vol. 34, no. 6, pp. 725-754, 2008.
[16] F. B. Ahmad, Z. Zhang, W. O. Doherty, and I. M. O'Hara, "The outlook of the production of advanced fuels and chemicals from integrated oil palm biomass biorefinery," Renewable and Sustainable Energy Reviews, vol. 109, pp. 386-411, 2019.
[17] E. Onoja, S. Chandren, F. I. Abdul Razak, N. A. Mahat, and R. A. Wahab, "Oil palm (Elaeis guineensis) biomass in Malaysia: the present and future prospects," Waste and Biomass Valorization, vol. 10, no. 8, pp. 2099-2117, 2019.
[18] H. B. Sharma, A. K. Sarmah, and B. Dubey, "Hydrothermal carbonization of renewable waste biomass for solid biofuel production: A discussion on process mechanism, the influence of process parameters, environmental performance and fuel properties of hydrochar," Renewable and sustainable energy reviews, vol. 123, p. 109761, 2020.
[19] M. Asadullah, A. M. Adi, N. Suhada, N. H. Malek, M. I. Saringat, and A. Azdarpour, "Optimization of palm kernel shell torrefaction to produce energy densified bio-coal," Energy Conversion and Management, vol. 88, pp. 1086-1093, 2014.
[20] A. A. Jaafar and M. M. Ahmad, "Torrefaction of Malaysian palm kernel shell into value-added solid fuels," International Journal of Chemical and Molecular Engineering, vol. 5, no. 12, pp. 1120-1123, 2011.
[21] U. Onochie, A. Obanor, S. Aliu, and O. Igbodaro, "Proximate and ultimate analysis of fuel pellets from oil palm residues," Nigerian Journal of Technology, vol. 36, no. 3, pp. 987-990, 2017.
[22] M. A. Sukiran, F. Abnisa, W. M. A. W. Daud, N. A. Bakar, and S. K. Loh, "A review of torrefaction of oil palm solid wastes for biofuel production," Energy Conversion and Management, vol. 149, pp. 101-120, 2017.
[23] G. Talero, S. Rincón, and A. Gómez, "Biomass torrefaction in a standard retort: A study on oil palm solid residues," Fuel, vol. 244, pp. 366-378, 2019.
[24] S. Kaewluan and S. Pipatmanomai, "Preliminary study of rubber wood chips gasification in a bubbling fluidised-bed reactor: effect of air to fuel ratio," in PSU-UNS International Conference on Engineering and Environment, 2007.
1.4 Comments 4: Line 53, “the energy costs resulting from electricity, fuel and water are 51, 43 and 6%, respectively.” 52 and 41 what? Percentage? Please specify.
Authors repose: 1.5 Comments 5: Too many abbreviations have been used in the manuscript. Thus, I highly recommend adding a nomenclature.
Authors repose: A nomenclature were added in manuscript follow reviewer suggestion, please see at line 30-73.
1.6 Comments 6: What is the rationale behind selecting the four conditions (88:12:0, 85:15:0, 85:13:2 and 85:10:5)? Why only four? Why not other values?
Authors repose: The author discussed with the boiler builder for thermal oil heaters (TOH) before starting the project. 100%w/w WC was used at the beginning of commissioning, but WC provides too much dust, causing operation to be stopped often for maintenance. After that, PKS was tried as a substitute for WC; the result got better but could not compete with the high production cost; then it was investigated to mix WC and PKS in different ratios until a ratio of 88: 12 %w/w gave a satisfactory result and continued to use with this formulation. The boiler team understood that, based on the manual, the boiler could only use WC and PKS, and the vendor requested only this. After brainstorming, the authors contacted other old and new TOH vendors and received updates from them. Old TOH can be modified and used with other biomass fuels such as OPMF, wood logs, and others. However, since glove manufacturing is located in southern Thailand, there are rubber wood and palm plantations; therefore, we are interested in exploring the use of WC; rubber wood residues, PKS and OMPF; wastes from oil palm mills. We start with the old formulation WC: PKS 88: 12 %w/w and try to blend it with OPMF according to the modified fuel funnel machine.
We compared the price and calorific value of three types of biomasses. The price per kg shows OPMF < WC < PKS while the heating value shows low price with high heating value compared to PKS > WCs > OPMF. WC is the central component of the mixture with others because they have been supplied sufficiently for a long time (the red color on the map is the area of rubber wood plantations), and the price is lower compared to PKS (the average price per ton in 2021 for WC and PKS is 32 USD and 74 USD, respectively), while the heating value is slightly lower (the HHV of WC and PKS is 16,850 KJ/kg and 17,154 KJ/kg, respectively). At the same time, OPMF receives the lowest price (26USD/ton), and the HHV (11,196KJ/kg) can reduce the production cost, but not enough to continue the supply as in WC. PKS and OMPF are waste from the oil palm mill, and the owner uses both for the boiler, and the rest for sale to another plant.
1.7 Comments 7: Lines 187-188, “OPMF biomass fuel showed the highest average moisture content of 42.71%, 187 followed by PKS (11.20%), while WC showed the lowest moisture content value of 9.76%.” what %? Vol. or wt.? Please specify clearly.
Authors repose: All of moisture content are % weight by weigh, the author had revised follow reviewer suggestion, please see at line 232-233.
1.8 Comments 8: Likewise, lines 253-254, “moisture 252 content of ≤8-12%” “Furthermore, the moisture content of raw biomass ranged between 14.8 and 41.8%” what %? Vol. or wt.? Please specify clearly.
Authors repose: All of moisture content are % weight by weigh, the author had revised follow reviewer suggestion, please see at line 297-298.
1.9 Comments 9: Same goes for line 257. What %? Vol. or wt.? Please specify clearly.
Authors repose: It’s %w/w please see at line 301.
1.10 Comments 10: What’s the meaning of using full form and abbreviations together repeatedly? For the first term, it is correct but later authors should use only abbreviations. Otherwise, remove abbreviations. For example, lines 280-281, lines 106-107 and lines 25-26.
Authors repose: The author had removed full form to correct, please see at line 322, 151 and 25.
1.11 Comments 11: This article lacks references. Only 14 references were provided. Many of them do not use proper reference styles. Authors must cite relevant references in due places.
Authors repose: The reference weas added 19 papers please see the detail at line 128-212.

Reviewer 3 Report
The study used wood chips, palm kernel shells, and oil palm mesocarp fibres as boiler fuel to produce rubber gloves. A total of four different ratios of WC, PKS, and OPMF were prepared and evaluated as boiler fuel. This manuscript should not be accepted for publication in Sustainability Journal in its current form. Following comments are provided to strengthen the manuscript. The authors are encouraged to resubmit the manuscript after carefully addressing the following comments.
· What is the main novelty of this work? Is there any similar work to compare?
· The authors should add pros and cons of biomass as boiler fuel in the Introduction section
· The language should be reviewed by a native speaker to improve the wording. There are many mistakes in this respect which should be resolved.
o For Example:
o Line 101: Consider the expression, ‘They used the existing formula for TOH unit and air velocity.”. To whom “they” is referred?
o Line 103: Used the correct form of the word “Ananlysed”
o Line 104: Correct form is “Materials and Methods”
o Lines 122-123: Consider the expression, ‘...and hydro statistic test pressure of...”. Modify the word “hydro statistic” to “Hydrostatic”.
o Line 130: Consider the expression, ‘...than OPMF, while OPMF while PKS had a lower...”. Two successive “while” are used improperly. Please correct it.
o Please check all over the manuscript and follow the comment.
· Lines 132-133: The abbreviations should be defined once, while the first time used in the manuscript. For Example: HHV, which should be defined in line 116 where it is used for the first time. Furthermore, the correct definition of the HHV is “Higher Heating Value”.
o Please check all over the manuscript and follow the comment.
· Equation (2): The mass flow rate units of measurement should be “kg/h” or “kg/s”. Correct “Delta temperature” to “Temperature difference”
· Equation (3):
o Correct “Temperature inlet” to “Inlet temperature”
o Correct “Temperature outlet” to “Outlet temperature”
· Equation (4):
o Correct “Qout” to “Qin”
· Lines 187-188: Discuss why biomass species are not dried before examination? Maybe the HHV of OPMF in dry form would be higher than the other biomass samples.
· Lines 194-195: The authors should discuss how concluded that moisture did not affect the calorific value and heat consumption in this study.
· Describe the experimental setup used for testing the biomass samples.
· The description of characterization and analysis techniques should be added to the “Materials and Methods” section.
· Figure 4: The authors shall improve the discussions not just describing the trends in the figure. Please check all over the manuscript and follow the comment. This is one of the critical concerns that should be resolved.
Author Response
Reviewer #3:
- The study used wood chips, palm kernel shells, and oil palm mesocarp fibres as boiler fuel to produce rubber gloves. A total of four different ratios of WC, PKS, and OPMF were prepared and evaluated as boiler fuel. This manuscript should not be accepted for publication in Sustainability Journal in its current form. Following comments are provided to strengthen the manuscript. The authors are encouraged to resubmit the manuscript after carefully addressing the following comments.
Authors repose: Thank you for your suggestion and I have corrected it accordingly.
1.1 What is the main novelty of this work? Is there any similar work to compare?
Authors repose: This paper addresses the use of fuel from multiple biomasses for glove production in TOH. From literature review the characteristics of single biomass fuel, life cycle analysis of glove production, economic and environmental value of biomass, performance and method to optimize TOH efficiency, biomass fuel combustion, and so on. However, there was no study in which a single biomass fuel was mixed, and the proximate and ultimate properties were analyzed, and the air emission after using biomass was not investigated at the same time. In addition, this study compared the production cost and economic efficiency of biomass fuel for glove production.
1.2 The authors should add pros and cons of biomass as boiler fuel in the Introduction section
Authors repose: Pros and cons of biomass as boiler fuel was added in the Introduction section, please see at line 128-193.
1.3 The language should be reviewed by a native speaker to improve the wording. There are many mistakes in this respect which should be resolved.
Authors repose: Thank you for your suggestions. Authors checked English by Professional proofreader.
For Example:
- Line 101: Consider the expression, ‘They used the existing formula for TOH unit and air ”. To whom “they” is referred?
Authors repose: Revised
- Line 103: Used the correct form of the word “Ananlysed”
Authors repose: Revised
- Line 104: Correct form is “Materials and Methods”
Authors repose: Revised
- Line 130: Consider the expression, ‘...than OPMF, while OPMF while PKS had a lower...”. Two successive “while” are used improperly. Please correct it.
Authors repose: Revised
- Please check all over the manuscript and follow the comment.
- Lines 132-133: The abbreviations should be defined once, while the first time used in the manuscript. For Example: HHV, which should be defined in line 116 where it is used for the first time. Furthermore, the correct definition of the HHV is “Higher Heating Value”.
Authors repose: The full name and abbreviations used for first time only, for repeat will using abbreviation. Revise version please see at line 174.
- Please check all over the manuscript and follow the comment.
Authors repose: All over the manuscript has been follow the comment.
Equation (2): The mass flow rate units of measurement should be “kg/h” or “kg/s”. Correct “Delta temperature” to “Temperature difference”
Authors repose: Revise version please see at line 197.
Equation (3):
3.1 Correct “Temperature inlet” to “Inlet temperature”
Authors repose: Revised
3.2 Correct “Temperature outlet” to “Outlet temperature” ·
Authors repose: Revised.
- Equation (4):
4.1 Correct “Qout” to “Qin”
Authors repose: Revised.
Lines 187-188: Discuss why biomass species are not dried before examination? Maybe the HHV of OPMF in dry form would be higher than the other biomass samples.
Authors repose: Mostly biomass fuel is using at boiler of glove manufacturing are not dry, the moisture depends on season and transportation. Entrepreneur has kept it at fuel storage and first-in first out using. Other studies do not clearly show that the HHV of OPMF is lower or higher than PKS and WC, please see the detail in table 2 (line 212).
- Lines 194-195: The authors should discuss how concluded that moisture did not affect the calorific value and heat consumption in this study.
Authors repose: Authors revised Information
- Describe the experimental setup used for testing the biomass samples.
Authors repose: Authors would like to thank for valuable comments. It has been added.
Analytical methods
Moisture content of biomass fuel : Gravimetric method/ ASTM D 3173
Proximate analysis : ASTM D 5468
Ultimate analysis
Moisture content : ASTM D 7582
Volatile matter : ASTM D 7582
Fixed carbon : ASTM D 7582
Ash value : ASTM D 3174
Gross heating value (As received basis) : Bomb Calorimeter/ AC-350/ BSEN 14918
Heat energy consumption : Read data from thermal hot oil machine/ and One-way ANOVA analysis
Moisture content of biomass fuel : Gravimetric method/ ASTM D 3173
Heat energy consumption : Read data from thermal hot oil machine/ and One-way ANOVA analysis
Production cost : One-way ANOVA analysis
Total suspension particulate : US.EPA Method 5
Sulfur dioxide : US.EPA Method 6C
Oxide of nitrogen : US.EPA Method 7E
Carbon monoxide : US.EPA Method 10
Carbon dioxide : US.EPA Method 3
- The description of characterization and analysis techniques should be added to the “Materials and Methods” section.
Authors response: Revised.
- Figure 4: The authors shall improve the discussions not just describing the trends in the figure. Please check all over the manuscript and follow the comment. This is one of the critical concerns that should be resolved.
Authors response: Revised.

Reviewer 4 Report
The article dealt with Biomass Fuel Ratio Optimization for Rubber Glove Manufacturing Cost Reduction in Thailand. I think that the paper is too superficial, the Introduction must be deeply rewritten, the modeling approach must be better defined and the results have to be interpreted more deeply. In my opinion, it is too much work for a major revision. Title should be improved to be technically sound
The novelty of the work should be further discussed.
A thorough general revision of the paper writing is needed to improve significantly the writing style.
I. Results from previous literature must be used for comparison and validation in the results section
II. The uncertainties of the key parameters should be discussed.
III. The Abstract must be deeply rewritten to reflect the originality of the paper
IV. The literature review should be deeply rewritten.
Author Response
- The article dealt with Biomass Fuel Ratio Optimization for Rubber Glove Manufacturing Cost Reduction in Thailand. I think that the paper is too superficial, the Introduction must be deeply rewritten, the modeling approach must be better defined, and the results have to be interpreted more deeply. In my opinion, it is too much work for a major revision. Title should be improved to be technically sound
Authors response: Thank you for your suggestion and I have corrected it accordingly. So that the research paper can be published to share with others for further use. Please note the details below.
- The novelty of the work should be further discussed.
Authors response: This paper addresses the use of fuel from multiple biomasses for glove production in TOH. Previous journals have studied the characteristics of single biomass fuel, life cycle analysis of glove production, economic and environmental value of biomass, performance and method to optimize TOH efficiency, biomass fuel combustion, and so on. However, there was no study in which a single biomass fuel was mixed, and the proximate and ultimate properties were analyzed, and the air emission after using biomass was not investigated at the same time. In addition, this study compared the production cost and economic efficiency of biomass fuel for glove production.
- A thorough general revision of the paper writing is needed to significantly improve the writing style.
Authors response: Manuscript has undergone English language editing by MDPI.
3.1 Results from previous literature must be used for comparison and validation in the results section
Authors response: Revised the result part.
3.2 The uncertainties of the key parameters should be discussed.
Authors response: All parameters are updated.
3.3 The Abstract must be deeply rewritten to reflect the originality of the paper
Authors response: Abstract is revised
3.4 The literature review should be deeply rewritten.
Authors response: The literature review has been rewritten, please see the detail at line 128-212.
Newly added:
Biomass is the third largest primary energy source after coal and oil [8][Bapat et al., 1997]. In addition, biomass fuels are known to be extremely environmentally friendly. First of all, biomass combustion does not produce CO2 gas because biomass absorbs atmospheric CO2 during growth for use in photosynthesis, which is equivalent to the amount released during combustion. Second, biomass burning also indirectly reduces the greenhouse gas CH4 by preventing the release of CH4 from biomass landfilling on agricultural land. In terms of global warming, CH4 has a 21 times stronger effect than CO2 [9][Sami et al., 2001]. Third, most biomass contains very little or no sulfur. Therefore, partial combustion of biomass from coal with high sulfur content can reduce SO2 gas content. In addition, combustion of high sulfur coal and alkaline biomass (i.e., K and Na ) in the biomass ash helps to capture some of the SO2 produced during combustion [10][Hein and Bemtgen, 1998]. Fourth, biomass fuels, such as wood and paper, contain less nitrogen than coal. In addition, biomass fuels release large amounts of NH3 during the evaporation process [11][Glarborg et al., 2003]. This ammonia helps convert NO to N2. Ammonia from biomass sources is considered a source of NOx reduction. Combining coal with biomass is a measure to reduce NOx from coal combustion. Finally, it helps to reduce soil and water pollution problems from landfills and biomass storage.
Total plant biomass consists of three main types of fibrous polymeric compounds: Cellulose, Hemicellulose, and Lignin[12] [Wiliams et al., 2001]. Lignocellulose The proportions of primary and secondary components of plants change. They depend on the species, type of plant tissue, age, and growing conditions. Biomass consists of the elements carbon (C), hydrogen (H), oxygen (O), nitrogen (N), sulfur (S), and chlorine (Cl). Ash from biomass contains the important elements Al, Ca, Fe, K, Mg, Na, P, Si, and Ti. The less important elements in the ash are AS, Be, Cd, Co, Cr, Cu, Hg, Mn, Mo, Ni, Pb, Sb, Tl, V, and Zn; of these elements in the ash, Ca and Mg led to an increase in the ash melting point, while K significantly lowered the ash melting point. Chloride compounds and alkali silicate compounds present in the ash lowered the ash melting point temperature [13, 14][Madhiyanon et al., 2012, 2013; Obernberger et al., 2006]. The weight percentages (dry) of C, H, and O in the biomass generally range from 30-60%, 5-6%, and 30-45%, respectively. N, S, and Cl are generally less than 1%. However, nitrogen content is sometimes higher because it is an essential food for plant growth. The properties of biomass fuels, which differ from those of coal fuels, affect the behavior of the combustion reaction. For example, the formation of ash particles on the surface of steam tubes and the emissions of biomass are different from those of coal [15][Yin et al., 2008]. The differences in the properties of the two fuels can be summarized as follows.
- Compared to coal fuels, biomass fuels generally contain more volatile components and oxygen. They have low carbon content and heating value.
- The pyrolysis process of biomass fuels starts at a lower temperature.
- The heat content of vaporization of biomass is about 70% compared to about 30% for coal.
- Biomass fuels such as rice straw and oil palm empty fruit branch contain more free alkali (K and Na, but mainly K) in the ash. This leads to more severe problems with ash melting or slagging and ash deposition on the heat exchanger surface or fouling than coal.
- Biomass charcoal has a better oxidation response than coal charcoal because it has a larger surface area and alkali is present in the charcoal as a catalyst [Blasi et al., 1999].
The behavior of biomass, which differs from coal, affects the use of biomass for thermal utilization and the selection of appropriate biomass combustion technology.
Several other factors may be even more critical when using biomass fuels for thermal power generation, including a sustainable supply of biomass. Some types of biomasses may only be available for a few weeks each year. Therefore, biomass must be accumulated for use throughout the year. This is different from fossil fuels. Some biomass must be prepared before it is transported to the combustor for further heat production, e.g., leaching, drying, or pelletizing.
Another drawback is the lack of technical data on how the fuel is transported into the combustion system, as well as technical data on the combustion characteristics and emissions of biomass fuels. In addition to the above obstacles, the private sector in Thailand still lacks appropriate and concrete incentive measures from the government. In addition, the nature of much potential biomass requires proper management from field collection to transportation to fuel processing. Until the fuel is transported to the boiler combustion chamber, the private sector, especially when investing in biomass fuel, does not have its fuel (e.g., rice mills with chaff or sugar mills that already have bagasse as a by-product) and must source fuel from outside sources. All these factors have caused the cost per unit of production to increase to the point where the investment may no longer be worthwhile. Therefore, in addition to rice husks, bagasse, palm fruit, and palm kernel shells, mills in Thailand rarely use biomass. Instead, bark (paper industry) is a by-product of the respective industry.
Biomass is the most popular alternative energy source because it is a clean, inexpensive, and widely available renewable resource. There are various biomasses, such as energy crops, wood, agricultural residues, municipal wastes, industrial wastes, etc., and there are various ways to convert biomass into energy. Incidentally, biomass is the only reliable resource that can be converted into all forms of energy and compression of biomass is the most important technique to get better properties in pellet form. In recent years, several studies have dealt with biomass pellets, such as wood waste and wood pellets, pellets from municipal solid waste, pellets from agricultural waste, from sewage sludge and pellets from industrial waste [17].
The palm oil industry produces significant amounts of solid waste. The solid wastes from the plantation are the oil palm trunk (OPT) and oil palm frond (OPF), while the processing plants generate empty fruit bunches (EFB), OPMF and PKS [16] [Ahmad et al., 2019]. Table 1 is summarizing the contents of lignocellulose in oil palm biomass and the proportions of cellulose, hemicellulose, lignin and ash in oil palm biomass. Therefore, proximate and ultimate analysis and heat value of raw biomass fuel is show in Table 2.
Table 1: The compositions of oil palm biomass.
Biomass type |
Cellulose |
Hemicellulose |
Lignin |
Ash |
References |
PKS |
28.8-27.2 |
21.6-22.7 |
44.0-50.7 |
8.6-16.3 |
[16]Ahmad et al., 2019 |
27.70 |
21.60 |
44.0 |
No data |
[17]Onoja et al., 2019 |
|
EFB |
34.0-40.4 |
17.2-22.4 |
23.1-29.6 |
5.0-6.5 |
[16]Ahmad et al., 2019 |
23.70 |
21.60 |
29.20 |
No data |
[17]Onoja et al., 2019 |
|
26.0 |
43.0 |
24.0 |
No data |
[17]Onoja et al., 2019 |
|
OPMF |
23.0-28.8 |
25.3-30.5 |
25.5-28.97 |
2.6-5.8 |
[16]Ahmad et al., 2019 |
19.0 |
37.0 |
33.0 |
No data |
[18]Sharma et al., 2020 |
|
OPF |
31.0-42.8 |
12.5-22.5 |
15.2-25.0 |
5.0-5.8 |
[16]Ahmad et al., 2019 |
OPT |
40.3-50.78 |
18.7-30.36 |
17.9-26.8 |
2.4-2.9 |
[16]Ahmad et al., 2019 |
Table 2: Proximate and ultimate analysis and heat value of raw biomass fuel.
Biomass |
Proximate analysis (wt, %) |
Ultimate analysis (wt, %) |
HHV |
LHV |
Reference |
|||||||
M |
FC |
VM |
Ash |
C |
H |
O |
N |
S |
(MJ/kg) |
(MJ/kg) |
||
PKS |
10.00 |
23.00 |
74.00 |
3.00 |
45.10 |
50.10 |
49.20 |
0.56 |
0.04 |
17.58 |
- |
[19]Asadullah et al. 2014 |
1.74 |
10.66 |
83.38 |
4.22 |
46.53 |
5.85 |
42.32 |
0.89 |
0.12 |
18.81 |
- |
[20]Jaafar and Ahmad 2011 |
|
11.00 |
19.70 |
67.20 |
2.10 |
49.74 |
5.32 |
44.86 |
0.08 |
0.16 |
16.30 |
- |
[17]Onoja et al., 2019; Ahmad et al., 2019 |
|
5.40 |
18.80 |
71.10 |
4.70 |
48.06 |
6.38 |
34.10 |
1.27 |
0.09 |
- |
- |
[17]Onoja et al., 2019 |
|
10.23 |
1.42 |
85.11 |
3.24 |
47.88 |
5.15 |
42.69 |
0.94 |
0.10 |
- |
- |
[21]Onochie et al., 2017 |
|
EFB |
66.00-69.00 |
10.80-14.50 |
86.50-87.70 |
3.70-5.30 |
48.72 |
7.86 |
48.18 |
0.25 |
- |
18.88 |
[22]Sukiran et al., 2017 |
|
8.78 |
8.60 |
79.65 |
3.00 |
48.79 |
7.33 |
40.18 |
n.d. |
0.68 |
16.80 |
- |
[17]Onoja et al., 2019; Ahmad et al., 2019 |
|
54.10-56.50 |
8.0-8.2 |
34.3-34.7 |
2.04-2.16 |
21.00-22.80 |
2.70-2.90 |
16.70-18.30 |
0.41-0.42 |
n.d. |
8.90-9.45 |
6.48-7.48 |
[23]Talero et al., 2019 |
|
n/a |
27.90 |
67.50 |
4.60 |
40.70 |
5.40 |
47.80 |
0.30 |
1.20 |
|
|
[24]Konsomboon et al., 2011 |
|
15.01 |
0.98 |
79.58 |
4.48 |
43.89 |
5.33 |
54.32 |
0.52 |
0.10 |
- |
- |
[21]Onochie et al., 2017 |
|
OPF |
62.00-77.00 |
3.20-14.80 |
83.60-88.30 |
3.20-3.80 |
48.43 |
10.48 |
46.50 |
12.40 |
- |
15.72 |
[22]Sukiran et al., 2017 |
|
OPT |
67.00-81.00 |
4.90-7.80 |
68.30-88.30 |
2.90-3.70 |
51.41 |
11.82 |
51.16 |
0.17 |
- |
17.47 |
[22]Sukiran et al., 2017 |
|
OPMF |
30.40-33.40 |
14.10-14.70 |
51.10-51.70 |
2.24-2.36 |
33.10-36.10 |
3.40-3.80 |
25.20-27.60 |
1.16-1.20 |
0.09 |
13.87-17.87 |
11.57-13.37 |
[23]Talero et al., 2019 |
n/a |
16.13 |
73.03 |
10.83 |
51.52 |
5.45 |
40.91 |
1.89 |
0.23 |
19.00 |
[17]Onoja et al., 2019; Ahmad et al., 2019 |
||
11.10 |
1.01 |
80.08 |
7.90 |
42.20 |
5.21 |
42.34 |
2.21 |
0.14 |
- |
- |
[21]Onochie et al., 2017 |
|
WC |
8.50 |
14.75 |
83.09 |
0.83 |
46.39 |
5.75 |
14.71 |
0.02 |
0.00 |
[24]and Pipatmanomai . 2007 |
||
6.40 |
3.90 |
15.30 |
74.5 |
19.30 |
4.60 |
22.40 |
0.10 |
0.10 |
9.13 |
[25]Sriket et al., 2019 |

Round 2
Reviewer 3 Report
Most of the comments have been addressed adequately by the authors. The experimental setup should be described and even a schematic diagram should be included in the manuscript, as suggested in comment #7 in the previous revision.
Author Response
The biomass was stored so that the moisture was below 50%. Weigh each biomass before mixing the recipe, and then the biomass loader is moved back and forth for at least 15 minutes to ensure that the mixture is homogeneous. The conveyor brings mixed biomass fuel (MBA, MBB, MBC, and MBD) to the TOH fuel chamber. The mixed biomass fuel is burned there, and the combustion heat leads to the oil heating system. This hot oil is fed to each area of the glove production line (cleaning tanks, coagulant dip tanks, latex dip tanks, pre- and post-leach tanks, and all ovens). The single and mixed biomass fuels sample is collected for proximate and ultimate analysis. Fuel consumption, heat consumption, and glove output were monitored, and each recipe was continued for three months. The wastes generated during combustion are ash and pollutant emissions. The ash can be disposed of in a landfill. The use of biomass ash has been investigated in many studies, such as road construction, cement industry, fertilizers, and bricks. Pollutant emissions were analyzed for this study. The emitted air pollution was collected from the stack of TOH. The process flow for biomass fuel mixing, feeding and heating supply to glove process has illustrated in Figure 3.

Reviewer 4 Report
The paper title should be changed to be technically sound.
Author Response
Thank you for your suggestion and I have revised it from “Biomass Fuel Ratio Optimization for Rubber Glove Manufacturing Cost Reduction in Thailand” to “Optimization of Biomass Fuel Composition for Rubber Glove Manufacturing in Thailand”
